# Cytomegalovirus Infection in Pregnancy Prevention and Treatment Options: A Systematic Review and Meta-Analysis

**DOI:** 10.3390/v15112142

**Published:** 2023-10-24

**Authors:** Magda Rybak-Krzyszkowska, Joanna Górecka, Hubert Huras, Magdalena Massalska-Wolska, Magdalena Staśkiewicz, Agnieszka Gach, Adrianna Kondracka, Jakub Staniczek, Wojciech Górczewski, Dariusz Borowski, Renata Jaczyńska, Mariusz Grzesiak, Waldemar Krzeszowski

**Affiliations:** 1Department of Obstetrics, Perinatology University Hospital, 30-551 Kraków, Poland; jsienicka@gmail.com (J.G.); staskiewicz.ma@gmail.com (M.S.); 2Hi-Gen Centrum Medyczne, 30-552 Kraków, Poland; 3Department of Obstetrics and Perinatology, Jagiellonian University Medical College, 30-551 Kraków, Poland; huberthuras@wp.pl; 4Clinical Department of Gynecological Endocrinology and Gynecology, University Hospital, 30-551 Kraków, Poland; mmassalska@su.krakow.pl; 5Department of Genetics, Polish Mother’s Memorial Hospital-Research Institute, 93-338 Łódź, Poland; a.gach@iczmp.edu.pl; 6Department of Obstetrics and Pathology of Pregnancy, Medical University of Lublin, 20-081 Lublin, Poland; adriannakondracka@wp.pl; 7Department of Gynecology, Obstetrics and Gynecological Oncology, Medical University of Silesia, 40-055 Katowice, Poland; jstaniczek@sum.edu.pl; 8Obstetrics and Gynecology Ward, Independent Public Health Care Facility “Bl. Marta Wiecka County Hospital”, 32-700 Bochnia, Poland; wojtekg1_9@op.pl; 9Clinic of Obstetrics and Gynecology, Provincial Combined Hospital in Kielce, 25-736 Kielce, Poland; darekborowski@gmail.com; 10Department of Obstetrics, Perinatology and Gynecology, Medical University of Warsaw, 02-091 Warsaw, Poland; renatajaczynska@tlen.pl; 11Department of Perinatology, Obstetrics and Gynecology, Polish Mother’s Memorial Hospital-Research Institute, 93-338 Łódź, Poland; mariusz.grzesiak@gmail.com (M.G.); waldemar.krzeszowski@gmail.com (W.K.); 12Department of Gynecology and Obstetrics, Medical University of Łódź, 93-338 Łódź, Poland; 13Salve Medica, 91-210 Lodz, Poland

**Keywords:** CMV, cytomegalovirus, congenital, pregnancy, infection, infants

## Abstract

Objectives: Cytomegalovirus (CMV) infection is a significant health concern affecting numerous expectant mothers across the globe. CMV is the leading cause of health problems and developmental delays among infected infants. Notably, this study examines CMV infection in pregnancy, its management, prevention mechanisms, and treatment options. Methods: Specifically, information from the Cochrane Library, PUBMED, Wiley Online, Science Direct, and Taylor Francis databases were reviewed along with additional records identified through the register, the Google Scholar search engine. Based on the search, 21 articles were identified for systematic review. Results: A total of six randomized controlled trials (RCTs) were utilized for a meta-analytic review. As heterogeneity was substantial, the random effects model was used for meta-analysis. Utilizing the random-effects model, the restricted maximum likelihood (REML) approach, the estimate of effect size (d = −0.479, 95% CI = −0.977 to 0.019, *p* = 0.060) suggests the results are not statistically significant, so it cannot be inferred that the prevention methods used were effective, despite an inverse relationship between treatment and number of infected cases. The findings indicated that several techniques are used to prevent, diagnose, and manage CMV infection during pregnancy, including proper hygiene, ultrasound examination (US), magnetic resonance imaging (MRI), amniocentesis, viremia, hyperimmunoglobulin (HIG), and valacyclovir (VACV). Conclusions: The current review has significant implications for addressing CMV infection in pregnancy. Specifically, it provides valuable findings on contemporary management interventions to prevent and treat CMV infection among expectant mothers. Therefore, it allows relevant stakeholders to address these critical health concerns and understand the effectiveness of the proposed prevention and treatment options.

## 1. Introduction

Cytomegalovirus (CMV) is the most common congenital infection worldwide. Annually, approximately 0.5–2.5% of newborns are born with congenital cytomegalovirus infection (cCMV), making the disease a global healthcare problem [1,2]. cCMV infection is the most typical cause of non-genetic hearing loss and neurologic disabilities, affecting 8–21% of infected children [3]. Even if children are born without signs of infection, about 20% will develop neurologic sequelae such as cerebral palsy, intellectual disability, epileptic seizures, and congenital, inflammatory eye disease [4]. In addition, 25 percent of these asymptomatic children develop late-onset hearing loss by the age of 4. cCMV may also be a significant contributor to antenatal stillbirth [5]. Therefore, the consequences of congenital CMV infection place a very substantial burden on health systems worldwide.

The virus is transmitted through direct or indirect contact with human secretions, such as urine, saliva, vaginal secretions, semen, breast milk, and blood products, and transplanted organs. Virus secretion is the longest in primary infection and it is the leading cause of congenital infection. As an immuno-incompetent organism, the fetus is particularly vulnerable to the consequences of intrauterine infection. Maternal–fetal transmission after primary maternal CMV infection occurs at a rate about 30% during first trimester of pregnancy, increasing over 70% at the third trimester [6]. The risk of health consequences for the fetus is highest when the virus is transmitted to the mother during the preconceptional period. Likewise, health challenges can occur if transmission occurs within the first trimester of pregnancy, reaching up to 30–40%, and decreases significantly with each trimester. Contrarily, the possibility of virus transmission via the placenta behaves inversely and is lowest in the first weeks of pregnancy and highest in the 3rd trimester. Therefore, determining the optimal prevention method is vital to reducing the risk of fetal infection. There is a limited number of worldwide guidelines for CMV screening in pregnant mothers; therefore, congenital diseases remain undetected in many asymptomatic children at birth until symptoms appear later in life [7,8,9]. This feature complicates the establishment of a clear link between disease symptoms and the verification of a congenital CMV infection. There is a growing knowledge pool of the mechanisms of fetal damage, expectations of pregnant patients, especially in the case of proven intrauterine infection, and the data from studies on antiviral drugs and immunoglobulins. Such developments necessitate addressing the possibility of prophylactic measures and pharmacological treatment, which can reduce adverse consequences for the fetus. Data on the use of antiviral drugs in pregnancy are growing. In addition, physicians have attempted immunomodulatory treatment with immunoglobulins.

The evidence favoring pharmacological interventions for CMV infections is increasingly growing. For women with compromised immune systems or organ transplant recipients, the approved antiviral drugs for the infection are ganciclovir, valacyclovir, cidofovir, and foscarnet [4]. An example of their use is when physicians use valacyclovir and gancyclovir for the treatment and prevention of congenital CMV infection. Valacyclovir acts against CMV’s DNA polymerase when utilized at a high dose. Shahar-Nissan et al.’s study revealed that valacyclovir effectively minimizes the rate of CMV infection among infants following primary maternal infection during early pregnancy. They argued that the early treatment of pregnant women might help avoid pregnancy termination or the delivery of infected infants [10]. Despite the outcome of this study on the efficacy of valacyclovir, there are no official CMV treatments during pregnancy. Likewise, effective measures to prevent maternal CMV infections and transmission to children are lacking. The existing guidelines recommend that physicians provide antenatal therapy as a research protocol treatment [7,8,9]. The vaccine to prevent this infection is unavailable, and treatment alternatives in pregnancy are limited. Pregnant women whose previous offspring attend day nurseries or kindergartens, nursery or kindergarten teachers, and medical staff in pediatric hospitals are at risk of contracting primary infections during pregnancy. Advances in CMV infection prevention and treatment are a priority globally. Literature data still indicate a low level of awareness of the risks associated with cCMV infection and the primary prevention options before and during pregnancy among both patients and healthcare providers [11,12,13,14,15]. Educational interventions and proper hygiene are effective measures to avoid CMV infection in pregnant women.

Caring for preschoolers while pregnant is a high risk; therefore, preventing maternal infection of CMV is necessary [1]. As with the treatment, there is no approved vaccine to prevent CMV infection among pregnant women. Health professionals recommend hygiene measures to avoid exchanging body fluids and prevent maternal contamination [1]. Prioritizing proper hygiene can help in preventing infection in pregnancy; education intervention is the most significant alternative strategy in minimizing the risk of the disease [4,16]. Enlightening expectant women on CMV infection and proper hygiene is a practical approach to preventing such infection among them.

The current review and meta-analysis highlight cytomegalovirus infection in pregnancy, covering its management and prevention options. The study prioritized articles discussing CMV infection among pregnant women. Given that CMV infection in pregnancy is an alarming problem, it is worth exploring ways to manage, prevent, and control it.

## 2. Methods

### 2.1. Eligibility Criteria

The preferred reporting items for systematic review and meta-analysis protocols (PRISMA-P) underpin the current review [17]. The protocol of this systematic review is registered in OSF Registries: https://doi.org/10.17605/OSF.IO/QFDRX (accessed on 14 August 2023). Studies associated with therapeutic interventions to manage, prevent, and treat congenital CMV infection in the population under investigation were included in the review. The focus in the selection of studies was on those investigating pregnant women and fetuses infected with CMV. Studies on general populations such as adult men and non-pregnant women were excluded from this review. Additionally, studies performed on patients with immunosuppression factors were omitted. This review prioritizes RCTs and cohort studies which were subsequently meta-analyzed. Review articles and case reports were excluded from the study. Finally, the study duration or publication date was limited to the years 2002–2022. The studies selected for the analysis were restricted to English, therefore, studies published in languages other than English were not considered in this review.

### 2.2. Information Sources

The primary information sources were from the following online literature databases: PUBMED, Wiley Online, Science Direct, Cochrane Library, and Taylor Francis Online, as well as sources of grey literature. Specifically, the Google scholar search engine was also used to access relevant and valuable information.

### 2.3. Search Strategy

The article search was conducted electronically on the following databases: PUBMED, Wiley Online, Science Direct, Cochrane Library, Taylor Francis Online. Applicable terms and keywords related to cytomegalovirus infection and pregnancy were used in this study. As for the grey literature, non-published studies, such as theses and dissertations, were analyzed to retrieve valuable information. The primary keywords included cytomegalovirus, CMV, pregnancy, and congenital.

### 2.4. Selection Process

The relevant studies were selected based on the inclusion and exclusion criteria. Articles written in English, full-text information, and relevant content based on their titles and abstracts were reviewed. The screening process was stepwise and followed an elaborate scanning procedure. The selected studies were screened, focusing on their abstracts and titles to determine their relevance or eligibility based on the inclusion criteria. Next, the full text of the selected studies was examined.

### 2.5. Data Collection Process

The review followed PRISMA guidelines [18].

### 2.6. Data Items

In the current review, the data items included pregnant women, therapeutic interventions to manage or prevent CMV infections, and treatment options for infection in pregnancy.

### 2.7. Study Risk of Bias Assessment

Assessing the risk of bias using the GRADE table will ensure that the selected studies are the most suitable for addressing the issue under review. Specifically, several domains of bias will be analyzed, including confounding factors, selection of participants in the study, intervention classification, missing information, and deviations from planned interventions. These domains will be classified based on the risk of biased judgment. Thus, the studies included in the review will be ranked following the overall bias risk.

### 2.8. Synthesis Methods

The selected studies for inclusion in this review were synthesized based on the type of intervention used. Specifically, various interventions were reported based on specificity, sensitivity analysis, and evaluation of the results measured concerning CMV infection. The study population was based on the diagnosis of CMV, including infection during pregnancy, abnormalities identified during an ultrasound scan, and infection at birth. During the synthesis of the findings, the primary outcomes involved the frequency of interventions for infection management during pregnancy. At the same time, the secondary outcomes were infants with or without cCMV, and with or without symptoms. The method of meta-analysis utilized was random-effects analysis using the Restricted Maximum Likelihood (REML) method [19]. As the variation was deemed important to be included in the uncertainty estimation of the regression coefficient estimates, fixed-effects models were ruled out as excess residual variance do not affect the computation of uncertainty estimates in fixed-effects models. The random-effects model generalizes the fixed-effect model, integrating components of variation between studies within the effect size, uncertain parameters, and estimates increasing residual variance [19].

## 3. Results

### 3.1. Study Selection

An electronic database and grey literature search generated 4074 articles after eliminating duplicates. Title or abstract screening resulted in the exclusion of 3997 journals that did not contain relevant information. In addition, the remaining 37 articles were screened on full-text review, leaving 21 studies that met the set inclusion criteria for the systematic review and 6 that met the criteria for the meta-analytic review. The excluded studies failed to meet the eligibility criteria. Researchers exclude studies for several reasons, including unsuitable study population or intervention. In this regard, 6 studies were included in the current systematic review and meta-analysis to explore CMV infection in pregnancy in terms of its management, prevention, and treatment options. The following PRISMA flowchart describes the study selection process for this review (Figure 1).

### 3.2. Study Characteristics

The studies had specific characteristics, including the author, year of publication, study design, and management interventions before and after prenatal diagnosis (Table 1).

### 3.3. Risk of Bias in Studies

As these were randomized studies, the risk of bias was considerably low for the meta-analysis. For the systematic review, the cohort studies carried a high risk for selection biases, performance bias, and detection bias (Table 2).

### 3.4. Results Synthesis

The key themes isolated based on an analysis of the RCT and cohort studies were prevention (primary and secondary), intervention, or management. The studies, overall, yielded important results. The synthesis of the literature review examined prevention utilizing hygiene as a form of primary prevention. Secondary prevention involved administration of hyper-immunoglobulin or valacyclovir prior to the confirmation of infection and before performing the amniocentesis. While secondary prevention carries a high risk of infection, it should be necessary to wait at least six weeks after presumed maternal infection to confirm or exclude maternal–fetal CMV transmission. Management and intervention, on the other hand, involves treatment with HIG and/or VACV, monitoring with ultrasound (US), and magnetic resonance imaging (MRI), and cordocentesis with platelet evaluation for inhibited hematopoiesis and impaired bone marrow.

### 3.5. Prevention

#### 3.5.1. Primary Prevention

Prevention of primary nature involves imparting hygiene information to mothers to prevent CMV in mothers and infants. A cohort study by Revello et al. involved imparting hygiene information to mothers in an intervention group (n = 331) as opposed to a control group (n = 315) who had no such intervention. Hygiene information was shared among pregnant women vulnerable to CMV infection due to occupational or personal reasons. The study involved testing CMV-seronegative mothers at the time of screening for maternal serum for fetus aneuploidy at 11 or 12 weeks of gestational age and giving hygiene recommendations before prosectively testing for CMV until delivery. While self-report measures were used to assess the acceptance of the hygiene recommendations as a secondary outcome, CMV seroconversion was the primary outcome. Researchers in the study found that in the intervention group (n = 381) who received hygiene counseling and CMV data, 13 women underwent miscarriage at 12–18 gestational weeks, and four of these women were not tested further. Another nine women remained CMV seronegative after a year following study completion. Around 37 women were not followed up, while 2 women seroconverted at 12–18 gestational weeks and 2 between 18 weeks and delivery. Only 4 (n = 331) women seroconverted after the completion of the study (1.2%, 95% CI 0.3–3.1). CMV-specific low avidity IgM and IgG in 5 women suggested detection of primary CMV during the first trimester after testing at 11–12 gestational weeks. In contrast, the comparison group involved 315 CMV seronegative women, wherein 24 mothers had undergone seroconversion (7.6%, 95% CI 4.9–11.1). Retrospectively, 4 additional women were diagnosed with primary CMV infection acquired in the first trimester of gestation due to CMV-specific IgM and IgG of low avidity. Therefore, the seroconversion rates were significantly lower for the intervention group than the control group (Δ = 6.4%, 95% CI 3.2–9.6, *p* < 0.001). Vulnerable pregnant women receiving CMV information showed lower risk,1.2:7.6, for acquiring CMV infection than uniformed mothers (crude odds ratio = 0.15, 95% CI 0.05–0.43). Regarding neonates infected in the intervention compared to the comparison group, of 28 neonates, 11 of them were infected in all, with 3 in the intervention group as opposed to 8 in the control group. Regarding the secondary outcome, hygiene information acquisition, 80 percent of the respondents followed recommendations either always (14%) or often (66%). A total of 93% of the respondents agreed that the CMV hygiene counseling and information was worth suggesting to mothers vulnerable to CMV infection. Therefore, the study effectively established that a primary prevention strategy could be effective in identifying and providing required CMV information and hygiene counseling to lower maternal CMV rates and congenital CMV infections [3].

#### 3.5.2. Secondary Prevention

Cohort studies and RCTs have established the efficacy of HIG, and CMV vaccinations, for preventing congenital CMV. Pass et al., in a phase 2 RCT study, evaluated the role of vaccine prevention strategies where three doses of recombinant CMV envelope glycoprotein E with MF59 adjuvant were compared with a placebo at three different times, namely 0, 1, and 6 months. In this RCT study, mothers were randomly assigned to either the treatment (n = 234) or control condition (n = 230) and after one year of follow-up, 49 confirmed infections were reported, with 18 in the vaccine group as opposed to 31 in the placebo group [22]. Within a 42-month period the vaccine group was more likely to be infection-free compared to the placebo group (*p* < 0.001) using Kaplan-Meier analysis. Pass et al. found infection rate estimates per 100 person years had a vaccine effectiveness rate of 50% (95% CI 7–73%). Additionally, one CMV case was noted among the vaccine group, as opposed to three infections in the placebo group. The study provided support for the use of a glycoprotein E for neutralizing antibodies, and such antibodies could play a powerful role in enhancing the protective effect of the vaccine [22].

Kagan et al. carried out a cohort study, where pregnant mothers (n = 40) were studied, with median gestational age at first presentation being an average of 9.6 weeks and ranging from 5.1 to 14.3 weeks. In the HIG treatment group, HIG doses were administered between one and six times for each patient. CMV-IgG showed periodic fluctuations on a bi-weekly period. In contrast, IgG avidity, after an initial increase after the first administration of HIG, remained stable throughout the treatment period. Maternal to fetal transmission before amniocentesis was noted in a single case, or 2.5%. Two additional subjects experienced late gestation transmission. All three cases with mother–fetus transmission brought transmission rates to 7.5% (95% CI 1.6–20.4%) in over 40 cases. Infected neonates showed no symptoms at birth. Matched historically, the control group covered 180 pregnant mothers. In this group, 38 transmissions (95% CI, 26.2–45.0%) took place, which was significantly higher (35.2%) than the HIG treatment group [31].

In another cohort study, Kagan et al. examined the effect of HIG on CMV prevention and detection. The study involved pregnant mothers (n = 149) and fetuses (n = 153), with median maternal age pegged at 32 years and weight at 65 kg. The median gestational age at referral was 9.4 weeks. Median anti-CMV IgG levels were 5.7 U/mL, while the anti-CMV IgM index was 2.5%, while 22.3% was the CMV IgG avidity. Amniocentesis was done at a median gestational age of 20.4 weeks. In 143 cases, or 93.5%, the fetuses were not infected, while 10 cases were marked by CMV maternal–fetal infection [38].

Nigro and Adler conducted a cohort study that also established the use of HIG as a CMV prevention therapy. Mothers (n = 304) and their infants (n = 281) participated in this study, where a follow-up was performed for 173 uninfected and 106 infected cases over four years. A total of 157 women were provided two doses of 200IU per kg of HIG per infusion. The researchers found an increase of 1.8 times the rate of congenital infection aOR = 5.2, *p* < 0.0001) and 1.8 times increase in maternal viral DNAemia prior to HIG administration (aOR = 3.0, *p* = 0.002). Abnormal ultrasounds were noted (aOR = 59, *p* = 0.0002). A diagnosis of material infection by seroconversion by avidity was present (aOR = 3.3, *p* = 0.007). The absence of HIG and abnormal ultrasounds were predictive of symptoms (*p* = 0.001). Long-term sequelae were predicted in those not receiving HIG (aOR = 13.2, *p* = 0.001). Abnormal ultrasounds were noted (OR = 7.6, *p* < 0.003) along with early maternal gestation (OR = 0.9, *p* = 0.017) [33].

In Devlieger et al.’s study, 4800 individuals were randomly assigned to the treatment arm, wherein 52 were seroconverted (median GA: 24 [11,12,13,14,15,16,17,18,19,20,21,22,23,24,25,26,27,28,29,30,31,32,33,34,35] weeks) and 45 were followed up. In this study, 4735 randomly chosen patients constituted the control group, from which 42 were seroconverted, 34 followed up (providing data for 28 infants), and 8 chose Cytotect off-label. In the control group cCMV rates were 13 out of 28 neonates (46.4% [CI 27.51–66.13]), as opposed to 16 out of 45 newborns (35.6% [CI 21.87–51.22]) (*p* = 0.46) in the treatment group [34].

Faure-Bardon et al., in their study, found that of 310 identified CMV-maternal primary infection (MPI) cases, 269 underwent amniocentesis, of which 65 accepted treatments using VACV. This longitudinal case–control cohort study serologically screened pregnant mothers between 2009 and 2020. For untreated cases, 65 mothers were selected as controls matched for periconceptional infections and gestational age at amniocentesis. Additionally, the researchers initiated VACV at 12.71 median gestational age and median duration of treatment being 35 days. Multivariate logistic regression showed that fetal infection in the treated group was lower (OR 0.318, 95% CI 0.120–0.841). Therefore, the acceptability, benefit, and tolerance of the VACV as a secondary prevention strategy was established [36].

In another related study by Faure-Bardon et al., the focus was on PCR by CVS as a means of evaluating the feasibility of the viral genome in the case of maternal primary infection with CMV during early pregnancy. In 37 pregnancies where CVS was performed, CMV PCR was positive in only 3 cases and negative in the remaining ones. CMV-PCR following amniocentesis at a median gestational range of 17.6 weeks was positive for the three cases which were also positive post CVS. Out of 34 patients with negative findings following CVS, amniocentesis excluded infection of the fetus in 31 cases, and confirmed it in the remaining three [35].

#### 3.5.3. Therapy of Fetal Infection

HIG has been widely used to manage CMV maternal fetal transmission post the contraction of CMV by the mothers. In a cohort study, Nigro et al. examined the effect of passive immunization on maternal–fetal CMV transmission. Of the 31 mothers in the study, one gave birth to an infant diagnosed with cCMV. HIG was linked to lower CMV incidences, with an adjusted odds ratio of 0.02, *p* < 0.001. Nigro et al. also utilized a control group of 14 mothers to assess the effect of the absence of HIG with 50% of the mothers (n = 7) not receiving HIG. Among the prevention group (n = 37 mothers) who received HIG, 6 of 37 (16%) had infants diagnosed with CMV than 19 of 47 mothers (40%) not receiving HIG. Specifically, HIG was associated with increased specific CMV-IgG concentrations, limited natural killer cells, lowered HLA DR + cells and raised avidity, signaling no adverse side effects [20]. Regarding the cohort studies, the key patterns of the results suggest that interventions, on average, were effective in preventing maternal–fetal transmission of CMV. Buxmann et al. administered prenatal CMV-HIG after the diagnosis of primary maternal CMV infection. A total of 115 doses were administered (n = 40 mothers and 6 fetuses). The treatment group comprised 4 mothers, while the multinomial group included 38 mothers and their 39 infants [23]. Intravenous CMV-HIG was administered in the treatment group under certain conditions. In the treatment group, three children emerged CMV positive, remaining asymptomatic at birth and at follow-ups, while one was symptomatic. In the multinomial group, 37 women and two infants received in utero CMV-HIG. In the multinomial group, 9 children in tested positive for cCMV, including TOP. In addition, 8 children in the multinomial group showed cCMV infection and were asymptomatic at birth and follow-up. No severe side effects occurred and HIG was tolerated well in those to whom it was administered. A smaller risk for intrauterine CMV maternal–fetus transmission was noted following the HIG administration. However, according to Buxmann et al., prenatal cCMV was not reversed after the HIG [23]. 

In contrast, a cohort study by Visentin et al. demonstrated ample proof of the positive impact of HIG in CMV prevention or for the associated adverse outcomes. Women with early primary CMV (n = 592) were part of the study, wherein 446 women were tested using amniocentesis and 92 fetuses were detected as having CMV, while 24 mothers underwent TOP, HIG was administered to 31 women. In contrast, a control group of 37 mothers received no treatment. Additionally, fetuses were matched on maternal age and the detection of pathology in the ultrasound and CMV load. Ultimately, 4 infants in the treatment condition (13%; 95% CI 1–25%) and 16 infants of untreated mothers (43%; 95% CI, 27–59%) presented with adverse outcomes (*p* < 0.001 by the 2-tailed Fisher exact test) [24].

Roxby et al. analyzed data for 141 infants (n = 148 mothers), compared results using ITT (intention to treat) analysis, and found differences between trial groups regarding susceptibility to CMV. Infant and mother characteristics at baseline were comparable. CMV infection rates between trial groups, however, did not differ significantly with 47 out of 71 newborns (66 percent) receiving valacyclovir (500 mg/twice a day) being infected and 46 as compared to 70 infants (66%) receiving a placebo diagnosed with CMV. The researchers did not note differences in CMV median time. In over 92% of the breast milk samples, CMV infection was discovered. The study, therefore, reported negative findings [25].

Revello et al.’s research revealed in the HIG (intervention) group, the rate of infant infection was 30% (18 fetuses, n = 61 mothers), but in the placebo group, it was 44% (27 fetuses, n = 62 mothers), highlighting the difference of around 14 percentage points (95% CI -3 to 31; *p* > 0.05) [26].

Nigro et al., in a cohort study of 351 mothers and 358 infants at a mean gestational birth age of 38.2 weeks, found multiple HIG doses of anywhere from one to eight were linked to increasing birth weight (*p* = 0.0006) and delivery-linked gestational age (*p* = 0.014). Infants without symptoms and those linked to multiple HIG maternal doses were associated with prevention of such fetal infections as CMV [27].

Delle Chiaie et al. retrospectively evaluated data from 50 females including three twin pregnancies. The mothers received 2 or more HIG infusions in the dose of 200 U/kg. The gestational median age at the time of CMV diagnosis in mothers was 13 weeks. Mothers received 142 doses of HIG, which were tolerated well. The pre-term birth rate was 4.2% in single pregnancies. CMV-linked sequelae in infants with cCMV was 10.5%. HIG application was favorably associated with a milder cCMV infection clinical course [28]. In contrast to the earlier finding by Buxmann et al., a milder course of the disease was noted following the administration of the HIG.

In contrast, in another cohort study, Minsart et al. administered CMV-specific HIG to16 women for suspected CMV during pregnancy while 55 mothers did not have any specific treatment and used bivariate analysis to show that HIG treatment did not considerably lower CMV or post-natal infections in the considerable treatment and prophylactic groups [29]. In contrast, a cohort study by Blazquez-Gamero et al. examined mother–infant pairs with a median gestation birth age of 39 weeks (interquartile range: 38–40) and two premature cases. Of 30 cases using amniocentesis, 21 fetuses showed a CMV PCR positive status (70%), one fetus with positive PCR results was injected an HIG dose and presented with CMV-PCR negative status showing no symptoms at 12 months. A total of 24 infants were infected at birth and 16 showed no CMV sequelae at 12 months. A clinical assessment was performed during the neonatal period and 12 months later in 16 and 15 infants, respectively. A total of 50% were symptomatic at birth and 4 in 15 showed losses of hearing at 12 months, while 3 were impaired neurologically. High sequelae of odds risk were noted for fetus before HIG administration, as suggested by ultrasonography. In contrast, in the prevention group, (n = 17), 7 fetuses, or 41%, were affected by CMV, while of 19 fetuses, 18 were diagnosed with cCMV and 8 showed CMV abnormalities in fetal ultrasonography before the HIG was administered [30]. In another cohort study, Seidel et al. also evaluated the role of HIG as a CMV management therapy. In a cohort of 46 mothers, 11 intrauterine infections were confirmed, which can be translated into transmission rate of 23.9%. In the control group, the transmission rate was 39.9%, marking a significant difference (*p* = 0.026). Hughes et al. demonstrated that the primary event outcome occurrence was noted in 46 fetuses or neonates (n = 203 mothers) (22.7%) in the hyperimmune globulin (intervention) group and in 37 fetuses (n = 191 mothers) (19.4%) in the placebo group (RR = 1.17; 95% CI 0.80–1.72; *p* > 0.05) [37]. A total of 10.3% and 5.4% infants were delivered with birthweight lower than 5th percentile in the intervention and control group, respectively (RR = 1.92; 95% CI 0.92–3.99) [32].

Another cohort study evaluated the effect of valacyclovir (8 g/day) on maternal–fetal transmission of CMV. Jacquemard et al. found that viral loads in fetal blood lowered following valaciclovir treatment within a span of one to 12 weeks (*p* = 0.02). A total of 20 pregnancies and 21 fetuses were treated at week 28, on average, spanning a range of 22–34 weeks for the duration, on average, of seven weeks (range 1–12 weeks). While 10 infants were developed normally between one and five years of age, six out of seven cases involved medical termination of pregnancy (TOP) due to cerebral lesions, while one fetus did not survive. In the control group not receiving valacyclovir treatment (n = 24), 14 or 58.3% had adverse outcomes such as TOP, CMV infection or fetal demise while the remaining 10 were healthy during the follow-up [21]. Shahar-Nissan et al., in their randomized double-blind placebo-controlled study, found that valacyclovir (2 × 4 g/day) therapy reduced fetal CMV infection rates after maternal primary infection early in pregnancy or periconceptual time. For the final analysis, 45 patients in the valacyclovir group and 45 in the placebo group were taken. The results indicated a 71% reduction in infection (11% in valacyclovir group vs. 30% in placebo; *p* = 0.027; OR 0.29, 95% CI 0.09–0.9) A majority of the RCTs showed the effectiveness of CMV prevention and treatment therapies such as valacyclovir and HIG, as well as passive vaccination [10].

### 3.6. Meta-Analysis

Utilizing the random effects model REML approach, the estimate of effect size (d = −0.479, 95% CI = −0.977–0.019, *p* = 0.060) suggests the results are not statistically significant, as shown in Table 3. Meta-analysis results suggested that overall conclusions based on the review of RCTs were not statistically significant. Two RCTs were outside the 95% confidence interval, suggesting that the results were not reliable for these studies, while the remaining 4 studies showed conflicting evidence. In addition, two studies suggested that the treatment group reported better CMV prevention and detection, while studies to the right of the effect line suggested that the outcome of interest occurred more frequently in the treatment or the control group.

### 3.7. Forest Plot

The study results show the effect estimates from individual studies, while diamonds demonstrate pooled results, as seen in Figure 2. Longer lines are associated with wider confidential intervals, pointing to low reliability of such study results, as noted in studies by Shahar-Nissan et al., Devlieger et al., and Revello et al. [10,26,34]. In contrast, the studies by Hughes et al., Pass et al. and Roxby et al. show higher reliability [22,25,37]. Studies to the left of the vertical line such as Pass et al. and Shahar-Nissan et al. showed that outcome of interest (CMV infection) occurred less frequently in the intervention than the control group, which Revello et al. also demonstrated to some extent [10,22,26]. In the study by Roxby et al., there were limited differences between the treatment and control group, demonstrating no particular effect of the CMV infection prevention intervention in the study [25]. On the other hand, Hughes et al. show the outcome of interest occurred more frequently in the treatment compared to the control group place on the right side of the vertical line [37]. The study by Devlieger et al. showed that the difference between the treatment and control group on the outcome of interest was not wide although CMV infections occurred less frequently in the intervention than the comparison or control group [34].

### 3.8. Publication Bias

The funnel plot demonstrates limited publication bias as the funnel plot is not completely asymmetrical, with two studies located on either side of the effect line and one study falling outside the 95% confidence interval on each side of the effect line, as seen in Figure 3 showing the funnel plot.

### 3.9. Sensitivity Analysis

The sensitivity analysis indicates that using a random-effects model produces disparate results compared to the fixed-effects model. As there was an inverse association between treatment and the number of infected patients, the results showed an estimate of −0.381 (95% CI = −0.662–0.101, *p* = 0.008), indicating that the preventative strategies used were effective. The results of the meta-analysis should be interpreted with caution.

### 3.10. Heterogeneity

The 1^2^ percentage or heterogeneity estimate suggests substantial heterogeneity at 64.83%. The studies are not homogenous, necessitating the use of the random effects model (Table 4).

## 4. Discussion

Primary CMV infection during pregnancies poses a significant risk of fetal transmission with the possibility of severe sequelae. Although there is no globally approved approach to prevent CMV infections, several mechanisms have been suggested and researched to determine their efficacy in preventing or treating these infections among expectant mothers. In other words, no suitable treatment option has been approved to address CMV infection in pregnancy. The current review and meta-analysis examined CMV infection in pregnancy, focusing on its prevention, management, and treatment options.

Early examination of the possibility of infection during prenatal diagnosis can help prevent cases of CMV in pregnancy. Amniocentesis is a method to withdraw amniotic fluid from the uterine cavity through a trans-abdominal approach using a needle [39]. Specifically, it is an invasive approach for the prenatal diagnosis of several pregnancy-related conditions [40]. The procedure can be done from at least 6–8 weeks post proven seroconversion or within 20–21 weeks of pregnancy otherwise. Amniocentesis allows for the confirmation or exclusion of the presence of the virus in the amniotic fluid and assess the viral load of CMV DNA in it [39]. Therefore, it is one of the practical approaches for detecting or excluding CMV infections during pregnancy.

HIG is also one of the approaches that have been suggested to prevent or minimize CMV infection vertical transmission. In other words, HIG is one of the interventions that can be utilized successfully in preventing maternal–fetal transmission. Furthermore, it has been proposed to treat primary CMV infection in pregnancy to minimize the risk of intrauterine viral transmission and morbidity associated with such a condition. Using HIG in pregnant women with primary CMV infection has proven safe and effective in preventing congenital disease using a high dosages. HIG use in pregnant women after CMV primary infection may not significantly reduce the infection rate; however, it is safe and can have favorable outcomes on the infected fetuses’ symptoms and sequelae. Accordingly, the risk of long-term sequelae in fetuses in the absence of US abnormalities before HIG is low, making it an appropriate option in handling infected fetuses. Detailed ultrasound evaluation of fetuses is recommended with special emphasis on looking for fetal infection-related changes in the placenta, fetal nervous system (detail fetal neurosonography assessment) and inflammation-related changes in other fetal organs. Likewise, valacyclovir is a potential congenital CMV preventive measure that helps to minimize vertical transmission. Specifically, it has been suggested as a practical approach in symptomatic CMV-infected fetuses to mitigate the risk of severe sequelae.

Additionally, based on the analysis of previous studies, the available treatment option for CMV infections during pregnancy include valacyclovir and HIG. US and MRI are valuable approaches to predicting the outcome of infected fetuses. They help in identifying lesions with poor prognoses. With the US technique, fetal abnormalities like intrauterine growth restrictions indicate infection [41]. The US scan method is non-invasive. In addition, it suggests symptomatic fetal infection, especially if the CMV infection has been confirmed by amniocentesis. Preventive treatment is effective in inhibiting viral transmission after maternal CMV infection. Specifically, this treatment is most effective, especially when started immediately after detecting presumed maternal contamination. Therefore, fetal transmission reduction is crucial among mothers infected during the early stages of their pregnancy.

Generally, the articles reviewed in the current study focused on the prevention and treatment options for CMV infection in pregnancy. The studies highlighted the various approaches and interventions to prevent or treat CMV infection among pregnant women. These techniques include HIG, serial monitoring, and valacyclovir. Despite the extensive research on the prevention and treatment of CMV infections among this population, there is no universally approved method to protect pregnant women from CMV infections. However, the meta-analysis also found these treatment and prevention strategies impacted the infection rate, in that there was an inverse relationship between treatment, prevention, and infection rates, as per the estimate of effect sizes, yet the results were not statistically significant.

Overall, the review focused on obtaining a global overview of the existing management, prevention, and treatment practices for CMV infections during pregnancy. However, the main limitation of this review is the heterogeneity in study design. Therefore, future reviews should focus on studies with homogeneous designs to yield the intended outcomes. The outcomes of this review will provide positive insights into the management of CMV infection; thus, significantly contributing to the existing body of knowledge.

## 5. Conclusions

The result of the systematic review suggests HIG as a form of CMV intervention for prevention and treatment of the viral maternal-to-fetal infection were effective, with cohort studies among the reviewed research showing HIG was linked to lower chances of CMV disease. HIG was associated with increased CMV-specific IgG concentrations, lowered natural killer cells, lowered HLA-DR + cells and raised avidity and normal development of infants following maternal CMV infections. Adverse outcomes, such as TOP, were avoided using HIG therapy.

However, while the systematic review suggests the treatment was effective when interventions such as hygiene prevention, valacyclovir and HIG were used, meta-analysis involving analysis of the effect of HIG and valacyclovir suggested mixed or inconclusive findings. The results of this study reveal the complex effect of HIG, valacyclovir, passive vaccination, and hygiene promotion as interventions for preventing or treating congenital CMV and preventing maternal–fetal viral transmission.

## Figures and Tables

**Figure 1 viruses-15-02142-f001:**
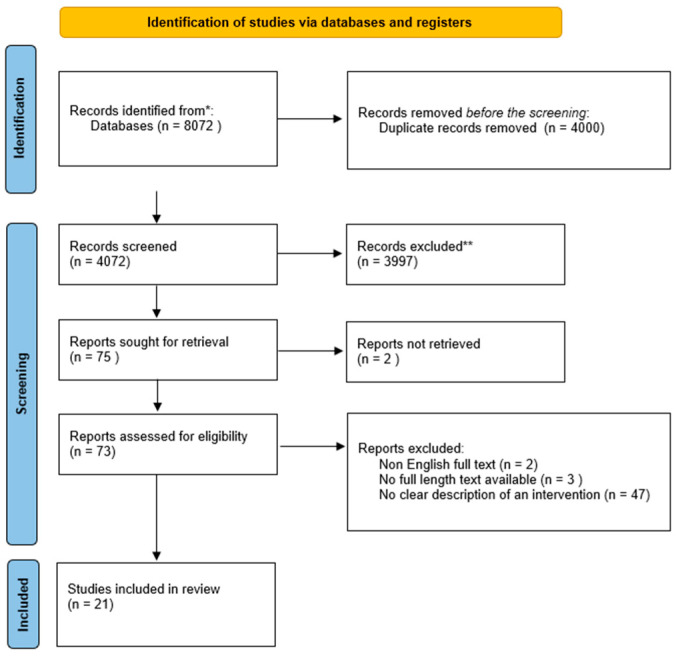
Systematic review flow chart. Note: * PUBMED, Wiley Online, Science Direct, Cochrane Library, Taylor Francis Online, and Google Scholar; ** out of the theme.

**Figure 2 viruses-15-02142-f002:**
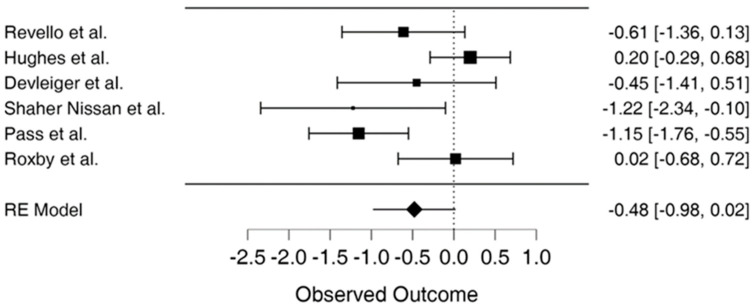
Forest plot for meta-analysis [3,10,22,25,34,37].

**Figure 3 viruses-15-02142-f003:**
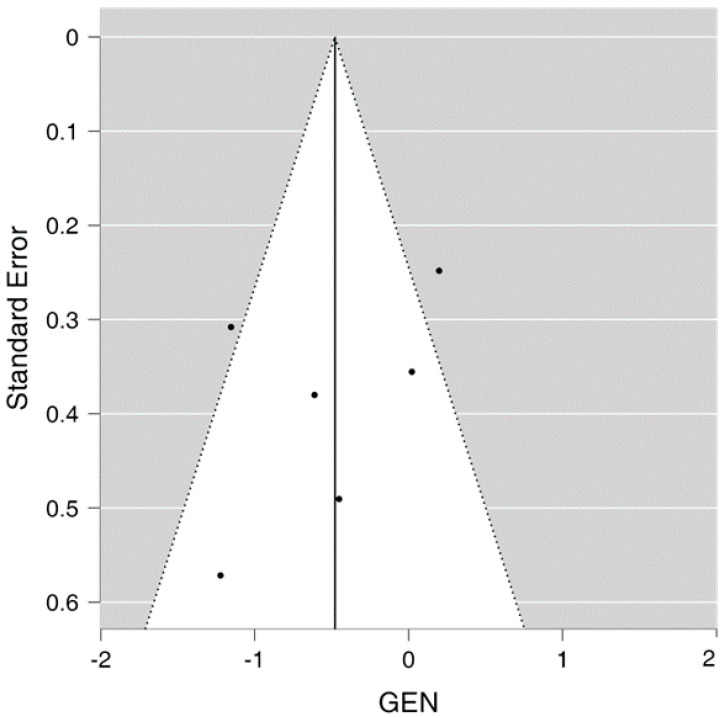
Funnel plot for the meta-analysis.

**Table 1 viruses-15-02142-t001:** Study characteristics.

Author(s)	Year	Design	Nature of Intervention	Post-Intervention Outcome
Nigro et al. [20]	2005	Cohort study	Passive immunization	Treatment group (n = 31): 31 mothers received HIG (200 IU/kg) after positive results of amniocentesis, 1/31 gave birth to an infant with cCMV. Control group (n = 14): 7/14 mothers who did not receive HIG after positive results of amniocentesis deliver symptomatic neonates Therefore, HIG was linked to lower chances of cCMV disease, (adjusted odds ratio: 0.02, *p* < 0.001). Prevention group (n = 37): 37 mothers who do not have amniocentesis received HIG (100 IU/kg) every month until delivery 0/37 women deliver symptomatic neonates Control group (n = 47) Women who do not have amniocentesis and not receive HIG, 3/47 women deliver neonates with severe symptoms of cCMV 6/37 (16%) had infants with cCMV, compared to 19/47 women (40%) not receiving HIG HIG increased CMV-specific IgG concentrations, decreased natural killer cells, raised avidity, and lowered HLA-DR+ cells with no adverse effects.
Jacquemard et al. [21]	2007	Cohort study	Valacyclovir (8 g/day)	Cohort (n = 21): Fetal blood viral load decreased after valacyclovir treatment within 1–12 weeks (Wilcoxon paired test *p* = 0.02) 20 women with 21 symptomatic CMV positive fetuses were treated at 28 weeks (range 22–34) for seven weeks (range: 1–12). Between 1 and 5 years of age, 10 infants developed normally. 6/7 cases required TOP (termination of pregnancy) due to cerebral lesions. One fetus did not survive. 38% rate of poor outcome in fetuses treated Valacyclovir (Fisher’s exact test *p* = 0.42). In the untreated group (n = 24), 14/24 (58.3%) had poor outcomes such as TOP (12 cases), fetal demise (1 case) or CMV inclusion disease (1 case), only 10 were healthy at follow-up.
Pass et al. [22]	2009	Phase II RCT	Vaccine prevention (CMV glycoprotein B vaccine)	Vaccine group (n = 178): 18/178 had CMV infection. Placebo group (n = 156): 31/156 had CMV infections. Vaccine group was more like to remain uninfected during 42 month period compering to placebo one (*p* = 0.02) Vaccine efficacy was 50% (95% CI, 7–73) based on infection rates per 100/person.
Buxmann et al. [23]	2012	Cohort study	HIG (200 IU/kg mother) Intraumbilical infusion (2 fetuses) Intraamniotic infusion (2 fetuses)	Cohort: n = 42 mothers and 43 children. Treatment group: n = 4 mothers and 4 fetuses. 3 of 4 mothers were administered HIG intravenously 3 children were CMV positive and did not show symptoms during follow-up or birth. 1 infant had cCMV infection in utero, during birth and follow-up. Multinomial group: n = 38 mothers and 39 infants. 37/38 mothers were intravenously administered HIG and 2 of 39 infants received HIG in utero. 9/39 children were positive for cCMV inclusive of a terminated pregnancy. All cases of cCMV showed no symptoms at follow-up or birth. Severe side effects were not noted in 115 CMV-HIG applications.
Visentin et al. [24]	2012	Cohort Study	HIG (200 IU/kg)	Cohort (n = 592): women with early primary CMV. 446 mothers had amniocentesis. 92 fetuses were CMV positive. 24 mothers terminated pregnancy. HIG was administered to 31. No treatment was received by 37. Fetuses of treated mothers and untreated mothers were matched on maternal age, CMV load, detection of pathological ultrasonography. 4/31 infants post a one-year evaluation for mothers who were treated (13%; 95% CI, 1–25%) and 16/37 untreated mothers (43%; 95% CI, 27–59%) showed adverse outcomes.
Roxby et al. [25]	2014	Phase II RCT	Valacyclovir (500 mg twice daily)	Study group (n = 71): 47/71 (66%) infected newborns. Control group (n = 70): 46/70 (66%) infected newborns.
Revello et al. [26]	2014	Phase II RCT	HIG (100 IU/kg)	Study group (n = 61): 33/61 had amniocentesis and 8/33 had positive results 18/61 had infections (transmission rate 30%) 3/25 neonates were positive at birth after negative results from amniocentesis. Control group (n = 62): 26/62 had amniocentesis and 10 had positive results There was none late transmission of the virus 27/62 had infections (transmission rate 44%) 14% increased risk (95% CI −3 to −31; *p* = 0.13).
Nigro et al. [27]	2015	Cohort study	HIG 100 IU/kg or 200 IU/kg (Italian cohort); 150–200 mg/kg (American cohort)	Cohort Study: 351 mothers, 358 infants. Mean gestational age at birth: 38.2 weeks. Presence of symptoms at birth. Multiple HIG doses ranging from 1–8 were associated with increasing birth weight (*p* = 0.0006) and gestational age at delivery (*p* = 0.014). All infants without symptoms and those who reported multiple maternal HIG doses were significantly associated with preventing fetal infections.
Revello et al. [3]	2015	Cohort study	Hygiene Information	Intervention Group (n = 331): 4/331 women seroconverted. Comparison group (n = 315): 24/315 women seroconverted. 3 babies in the intervention group were infected with CMV compared to 8 in the comparison group.
Delle Chiaie et al. [28]	2018	Cohort study	HIG (200 IU/kg)	Cohort (n = 50 women, 53 neonates): Median gestation age at maternal CMV diagnosis = 13 weeks 142 material HIG doses were given. No HIG adverse side effect was noted. 19/53 fetuses had cCMV diagnosed prenatally 2/19 neonates were symptomatic at birth Frequency of CMV related sequelae in infants with cCMV infection was 10.5%.
Minsart et al. [29]	2018	Cohort Study	HIG (150 mg/kg)	Cohort (n = 71): 16/71 received HIG 55/71 had no CMV specific treatment. Cytomegalovirus specific hyperimmunoglobulins (HIG) treatment was well tolerated. Bivariate analysis showed HIG treatment did not significantly decrease CMV or postnatal infections in both treatment and prophylactic groups.
Blazquez-Gamero et al. [30]	2019	Cohort study	HIG (100 IU/kg monthly in PG, 200 IU/kg in TG	Cohort: 36 women, median gestational age at birth 39 weeks (IQR: 38–40) and two premature cases. Of 30 cases had Amniocentesis, CMV PCR was positive in 21 (70%). Prevention Group-PG (N = 17): 6/16 (37.5%) fetuses were infected at birth,1/17 TOP 1/17 abnormalities in fetal US. Birth: hearing loss 1/6 (16.7%), motor impairment 0/6 (0%); symptomatic 1/6 (16.7%). 12 months of life: hearing loss 1/6 (16.7%), neurologic abnormalities 0/6 (0%). Treatment Group-TG (n = 19): 18/19 fetuses were infected at birth (95%). 8/19 fetuses showed CMV abnormalities in fetal US before HIG treatment. Birth: hearing loss 4/16 (25%), motor impairment 3/16 (18.8), symptomatic 8/16 (50%), 2 cases without clinical data and lost follow up. 12 months of life: hearing loss and neurologic abnormalities 3/15 (20%). Hearing loss 1/15 (6.7%), 1 case lost follow up.
Kagan et al. [31]	2019	Cohort Study	HIG (200 IU/kg biweekly until 200 weeks of gestation)	Cohort (n = 40) receive HIG: Minimum HIG number of doses: 2, maximum of 6. Mother to fetus transmission before amniocentesis was noted in 1/40 (2.5%). 2/39 (5.12%) had late gestation transmission. Transmission rate 7.5% (95% CI, 1.6–20.4%) Infected neonates showed no symptoms at birth. Matched historical Control group (n = 108): 38/108 (35.2%) took place in the control group (95% CI, 26.2–45%, *p* < 0.001).
Seidel et al. [32]	2020	Cohort Study	HIG (dose 200 IU/kg, two or more infusion in 2-weeks interval)	Cohort group (n = 46): 11 cases of maternal–fetal transmissions of infection (transmission rate 23.9%) Match random Control group (n = 82): (Transmission rate of 39.9%) significant reduction risk (*p* = 0.026)
Nigro and Adler [33]	2020	Cohort study	HIG (200 IU/kg)	n = 304 women and 281 infants Follow-up was carried out for 106 infected and 173 uninfected fetuses at 4 years. 157 women were given 2 doses HIG. Fetal/neonatal infection was marked by the following factors: 1.8-fold increase in the congenital infection rate without HIG (adjusted odds ratio (AOR) 5.2, *p* < 0.0001). 1.8-fold increase linked to maternal viral DNAemia before HIG administration (AOR 3.0, *p* = 0.002). Abnormal ultrasounds (AOR 59, *p* = 0.0002). Diagnosis of material infection by seroconversion rather by avidity (AOR 3.3, *p* = 0.007). Lack of HIG and abnormal ultrasounds also predicted symptoms (AOR 59, *p* = 0.001) Symptoms and long-term sequelae were marked by: Long-term sequelae were predicted by not received HIG (AOR 13.2, *p* = 0.001). Abnormal ultrasounds findings (OR 7.6, *p* < 0.003). Early gestation maternal infection (OR 0.9, *p* < 0.017).
Devlieger et al. [34]	2021	Phase III RCT	Serial monitoring and HIG (200 IU/kg twice be-weekly)	Treatment group (n = 45 completed follow up): 16/45 newborns with cCMV (35.6%). At birth: US anomalies 4/16 (25%), hearing loss 1/16 (6.7%). Control group (28/35 completed follow up): 13/28 newborns with cCMV (46.4%). At birth: US anomalies 2/11 (18.2%), hearing loss 0/10/ 23% relative reduction in cCMV (*p* = 0.46, Fisher’s exact).
Shahar Nissan et al. [10]	2020	Phase III RCT	Valacyclovir (8 g/day)	Study group (n = 45): 5/45 (11%) CMV DNA positive amniocentesis. Control group (n = 47): 14/47 (30%) CMV DNA positive amniocentesis. 71% reduction in infection (*p* = 0.027; OR 0.29, 95% CI 0.09–0.9)
Faure-Bardon et al. [35]	2021	Cohort study	“Amplification of the viral genome by polymerase chain reaction (PCR) analysis of trophoblast samples obtained by chorionic villus sampling”	CMV-PCR positive in 3 cases and negative in 34 cases (n = 37). Amniocentesis was positive for 3 cases and negative in 31 cases (n = 34 CVS-PCR negative patients). Sensitivity analysis = 50% (95% CI 19–81). Specificity = 100% (95% CI 89–100%). Positive predictive value = 100% (95% CI 44–100%). Negative predictive value = 91% (95% CI 77–97%).
Faure-Bardon et al. [36]	2021	Cohort study	Valacyclovir (8 g/day)	Of 310 cases, 269 underwent amniocentesis for PCR. 65/269 accepted treatment with VACV at median gestational age of 12.71 (IQR, 10.00–13.86). Median duration of treatment was 35 days (IQR 26–54). Fetal infection was lower in treated group 8/65 (12%) compared to the historical group without the treatment 19/65 (29%) (*p* = 0.029) and risk of transmission decreased significantly. (OR = 0.318, 95% CI = 0.120–0.841, *p* = 0.021). A patient experienced acute renal failure four weeks post VACV therapy was initiated until treatment ceased.
Hughes et al. [37]	2021	Phase II RCT	HIG (100 mg/kg monthly until delivery)	Treatment group (n = 203): 46/203 (22.7%) infections. Placebo group (n = 191): 37/191 (19.4%) infections. Primary outcome was composited: cCMV in fetal or neonatal period up to 21 week of life, fetal or neonatal death inc. TOP without fetal or neonatal testing toward CMV infection. RR 1.17 (95% CI, 0.8–1.72, *p* = 0.42)
Kagan et al. [38]	2021	Cohort Study	HIG (200 IU/kg be weekly until 18 weeks of gestation)	Cohort: n = 149 pregnancies (153 fetuses). Median anti CMV IgG level = 5.7 U/mL. Anti CMV-IgM Index = 2.5%. CMV IgG avidity = 22.3%. HIG treatment at median gestational age = 20.4 weeks 4 doses average. 143/153 fetuses (93.5%) were not infected. 10/153 fetuses (6.5%) were infected. (6.5% (95% CI, 3.2–11.7%)) 2 newborns were positive CMV and asymptomatic at birth.

**Table 2 viruses-15-02142-t002:** GRADE–the risk of bias.

STUDY	I	II	III	IV	V	VI	VII
Nigro et al. (2005) [20]							
Jacquemard et al. (2007) [21]							
Pass et al. (2009) [22]							
Buxmann et al. (2012) [23]							
Visentin et al. (2012) [24]							
Roxby et al. (2014) [25]							
Revello et al. (2014) [26]							
Nigro et al. (2015) [27]							
Revello et al. (2015) [3]							
Delle Chiaie et al. (2018) [28]							
Minsart et al. (2018) [29]							
Blazquez-Gamero et al. (2019) [30]							
Kagan et al. (2019) [31]							
Seidel et al. (2020) [32]							
Nigro and Adler (2020) [33]							
Devlieger et al. (2020) [34]							
Shaher Nissan et al. (2020) [10]							
Faure-Bardon et al. (2021) [35]							
Faure-Bardon et al. (2021) [36]							
Hughes et al. (2021) [37]							
Kagan et al. (2021) [38]							

I is random sequence generation (selection bias); II is allocation concealment (selection bias); III is blinding of participants and personnel (performance bias); IV is blinding of outcome assessment (detection bias); V is incomplete outcome data (attrition bias); VI is selective reporting (reporting bias); VII is other bias. High risk 

, Moderate risk 

, Low risk 

.

**Table 3 viruses-15-02142-t003:** Random Effects Model REML approach.

	Q	df	*p*
Omnibus test of Model Coefficients	3.548	1	0.060
Test of Residual Heterogeneity	15.571	5	0.008

Note. *p*-values are approximate. The model was estimated using the Restricted ML method.

**Table 4 viruses-15-02142-t004:** Residual Heterogeneity Estimates.

	95% Confidence Interval		
	Estimate	Lower	Upper
τ^2^	0.240	0.019	1.913
τ	0.490	0.140	1.383
I^2^ (%)	64.829	12.989	93.618
H^2^	2.843	1.149	15.670

## Data Availability

The data are available upon the reasonable request.

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
