# Peer review of "Cytomegalovirus Infection in Pregnancy Prevention and Treatment Options: A Systematic Review and Meta-Analysis"

_viruses, 2023, doi:10.3390/v15112142_

Round 1
Reviewer 1 Report
Interesting article, difficult to read, in paragraph 3.5.1 in the description of Revello study there are some discrepant numbers
Even in the other paragraphs the descriptio of the studies is difficult to follow
Author Response
Dear Reviewer,
Thank you for reviewing our manuscript and giving suggestions. We have corrected our manuscript. Please see point-by-point answers below:
Interesting article, difficult to read, in paragraph 3.5.1 in the description of Revello study there are some discrepant numbers
This paragraph has been corrected.
Even in the other paragraphs the descriptio of the studies is difficult to follow
Other paragraphs have been corrected too. We hope the manuscript is easier to read now.
Best wishes,
Reviewer 2 Report
The article under consideration presents a well-structured and eloquently written account of research related to cytomegalovirus (CMV).
The usage of the English language in this article is of a remarkably high standard, facilitating clear communication of complex scientific concepts.
When you first mention a term that you plan to abbreviate, provide its complete definition or full name, followed by the abbreviation in parentheses.
The introduction mentions the lack of worldwide guidelines for CMV screening of pregnant mothers. Are there any regional or national guidelines that have been developed, and if so, how effective have they been in reducing congenital CMV infections?
Figure 1 could be enhanced by incorporating details regarding the number of articles retrieved from each database and the count of articles excluded based on different exclusion criteria.
In the synthesis methods section, you mentioned using the random-effects model with the Restricted Maximum Likelihood (REML) method for meta-analysis. Could you elaborate on how this method was applied and why it was chosen over other meta-analysis approaches?
It's essential to discuss any potential limitations or biases in these studies that might impact the interpretation of the results, such as the study design, sample size, and potential confounding factors.
Given the substantial heterogeneity (64.83%), it's essential to explore potential sources of heterogeneity among the included studies. Did the authors investigate factors such as differences in study populations, interventions, or study designs that might contribute to this heterogeneity?
Author Response
Dear Reviewer,
Thank you for reviewing our manuscript and giving suggestions. We have corrected the manuscript. Please see point-by-point response below:
The article under consideration presents a well-structured and eloquently written account of research related to cytomegalovirus (CMV).
The usage of the English language in this article is of a remarkably high standard, facilitating clear communication of complex scientific concepts.
When you first mention a term that you plan to abbreviate, provide its complete definition or full name, followed by the abbreviation in parentheses.
We have corrected abbreviations.
The introduction mentions the lack of worldwide guidelines for CMV screening of pregnant mothers. Are there any regional or national guidelines that have been developed, and if so, how effective have they been in reducing congenital CMV infections?
The national and regional guidelines have been added.
Figure 1 could be enhanced by incorporating details regarding the number of articles retrieved from each database and the count of articles excluded based on different exclusion criteria.
Unfortunetly, we do not have this information stored.
In the synthesis methods section, you mentioned using the random-effects model with the Restricted Maximum Likelihood (REML) method for meta-analysis. Could you elaborate on how this method was applied and why it was chosen over other meta-analysis approaches?
It's essential to discuss any potential limitations or biases in these studies that might impact the interpretation of the results, such as the study design, sample size, and potential confounding factors.
Certainly, the choice of using the random-effects model with the Restricted Maximum Likelihood (REML) method for meta-analysis is based on several considerations.
REML is a statistical estimation method used to estimate the parameters of a random-effects model.
It is considered superior to other methods like the Method of Moments (MOM) for estimating the variance components of the random-effects model because it provides unbiased and efficient estimates.
Here's how the REML method is typically applied in the context of meta-analysis:
Data Collection: Gather relevant studies that report effect sizes (e.g., mean differences, odds ratios, hazard ratios) and their associated standard errors or confidence intervals. The studies should address a common research question or hypothesis.
Estimation of Parameters: Use the REML method to estimate the parameters of the model. Specifically, REML estimates the overall mean (μ) and the between-study variance, which quantifies the degree of heterogeneity among the studies.
Meta-Analysis: Calculate the overall effect size and its confidence interval based on the estimated parameters. This provides a summary estimate of the effect size across all included studies, accounting for both within-study error and between-study variability.
In summary, the choice of the random-effects model with REML in meta-analysis is driven by the need to account for heterogeneity among studies and to obtain robust and unbiased estimates of the overall effect size.
Given the substantial heterogeneity (64.83%), it's essential to explore potential sources of heterogeneity among the included studies. Did the authors investigate factors such as differences in study populations, interventions, or study designs that might contribute to this heterogeneity?
Yes, we have identified potential sources of heterogeneity:
- Studies include participants with varying demographic characteristics, clinical profiles, or comorbidities.
- Variation in method of treatments, dosages as well as treatment durations
- Differences in study designs (randomized controlled trials and observational studies)
- Differences in healthcare practices, standards of care, or patient populations across geographic regions or time periods
This is why we used the REML method in the meta-analysis, which is most responsive when high heterogeneity is found (as described above).
We hope our manuscript will be accurate for publication in this form.
Best wishes